# Impact of expanding smoke-free policies beyond enclosed public places and workplaces on children's tobacco smoke exposure and respiratory health: protocol for a systematic review and meta-analysis

Márta K Radó ⬚ ,[1,2] Famke JM Mölenberg ⬚ ,[2] Aziz Sheikh,[3,4] Christopher Millett ⬚ ,[5] Wichor M Bramer ⬚ ,[6] Alex Burdorf ⬚ ,[2] Frank J van Lenthe,[2] Jasper V Been ⬚ [1,2]

For numbered affiliations see end of article.

**Correspondence to**
Dr Jasper V Been;
j.been@erasmusmc.nl

## ABSTRACT

**Introduction** Tobacco smoke exposure (TSE) has considerable adverse respiratory health impact among children. Smoke-free policies covering enclosed public places are known to reduce child TSE and benefit child health. An increasing number of jurisdictions are now expanding smoke-free policies to also cover outdoor areas and/or (semi)private spaces (indoor and/or outdoor). We aim to systematically review the evidence on the impact of these 'novel smoke-free policies' on children's TSE and respiratory health.

**Methods and analysis** 13 electronic databases will be searched by two independent reviewers for eligible studies. We will consult experts from the field and hand-search references and citations to identify additional published and unpublished studies. Study designs recommended by the Cochrane Effective Practice and Organisation of Care (EPOC) group are eligible, without restrictions on the observational period, publication date or language. Our primary outcomes are: self-reported or parental-reported TSE in places covered by the policy; unplanned hospital attendance for wheezing/asthma and unplanned hospital attendance for respiratory infections. We will assess risk of bias of individual studies following the EPOC or Risk Of Bias In Non-randomised Studies of Interventions tool, as appropriate. We will conduct separate random effects meta-analyses for smoke-free policies covering (1) indoor private places, (2) indoor semiprivate places, (3) outdoor (semi)private places and (4) outdoor public places. We will assess whether the policies were associated with changes in TSE in other locations (eg, displacement). Subgroup analyses will be conducted based on country income classification (ie, high, middle or low income) and by socioeconomic status. Sensitivity analyses will be undertaken via broadening our study design eligibility criteria (ie, including non-EPOC designs) or via excluding studies with a high risk of bias. This review will inform policymakers regarding the implementation of extended smoke-free policies to safeguard children's health.

## Strengths and limitations of this study

► This systematic review will be the first to comprehensively synthesise the evidence regarding the impact of 'novel smoke-free policies' (ie, those covering (1) indoor private places (2) indoor semiprivate places (3) outdoor (semi)private places and (4) outdoor public places) on tobacco smoke exposure and health outcomes among children.

► This protocol has been designed in line with the Preferred Reporting Items for Systematic Reviews and Meta-Analyses for Protocols guidelines and guidelines of the Cochrane Effective Practice and Organisation of Care.

► A possible limitation is that included studies might be heterogeneous in the policy, study design and data collection methods, which might limit our ability to synthesise the results using meta-analysis.

► The value of this systematic review depends on the quality and availability of the evidence on the topic.

► This work will be instrumental in informing effective policy-making regarding implementing novel smoke-free policies to protect children from the adverse effects of tobacco smoke.

**Ethics and dissemination** Ethical approval is not required. Findings will be disseminated to academics and the general public.
**PROSPERO registration number** CRD42020190563.

## INTRODUCTION

Tobacco smoke exposure (TSE) is an important cause of adverse respiratory health outcomes in children worldwide. TSE includes both second-hand smoke (SHS) and third-hand smoke (THS) exposure. SHS refers to the inhalation of emitted smoke, while THS

exposure refers to the uptake of tobacco smoke residuals from polluted surfaces after someone has finished smoking. TSE is estimated to cause 166 000 child deaths annually associated with an increased risk of respiratory tract infections (RTIs), wheezing and asthma.[1 2]

Safeguarding of child health provides justification to advocate for strong tobacco control measures.[3] Previous systematic reviews found that smoke-free policies in enclosed public places are successful in reducing adverse child health outcomes. Meta-analyses showed that these policies were followed by a 9.8% (95% CI 3.0% to 16.6%) reduction in hospital attendance for asthma exacerbations, and an 18.5% (95% CI 4.2% to 32.8%) reduction in hospital attendance for lower RTIs.[2] These health impacts of smoke-free policies are likely mediated through a reduction in SHS and potentially also in THS.[1] Indeed, individual studies have demonstrated that smoke-free policies covering enclosed public areas are successful in decreasing child TSE in public places and even in private places, such as cars or homes, most likely via changes in social norms.[4–8] Although in low-income and middle-income countries high background air pollution, poor economic conditions and low awareness of tobacco-related harm might obscure the positive effects of smoke-free legislation, evidence suggests that smoke-free policies can have a similarly positive impact in these countries as in high-income countries.[9–11]

As a result, smoke-free legislation is increasingly recognised as an important policy tool to protect children from the adverse health effects of TSE (eg, incorporated in Sustainable Development Goals 3.2, 3.4, 3.9 and 3.A that aim to improve health and well-being).[3] Recently, policies to provide additional protection of children to TSE beyond enclosed public places have been implemented in a number of places, both at a national and a subnational level. These include expansion to cover outdoor areas frequented by children such as school grounds, playgrounds and parks.[12] Also, some countries and subnational regions have now implemented policies prohibiting smoking in semiprivate spaces such as shared housing or private spaces such as cars.[13 14]

The positive effects of smoke-free policies covering enclosed public places on child health are now well established.[2 15] A systematic synthesis of the evidence on the effectiveness of smoke-free policies covering outdoor and/or (semi)private spaces to reduce children's TSE and benefit their health is however currently lacking.[2 15] It is important to note that effectiveness of smoke-free policies covering indoor public places cannot be easily extrapolated to those covering outdoor or private spaces, due to various reasons (eg, dilution of TSE in outdoor spaces, enforcement issues). Further, evidence from various countries indicates that smoke-free policies covering public enclosed places and workplaces are followed by reductions in TSE even in areas not covered by the policy through norm spreading. Whether novel smoke-free policies also have an impact on TSE in places not covered by the policy is unclear. Theoretically, there may be a

reduction (via norm spreading) in such places but there may also be displacement of smoking to areas not covered by the policy.[14] Finally, effectiveness of smoke-free policies is likely to vary according to local rates of smoking and children's SHS exposure, and comprehensiveness of the policy as well as compliance and enforcement.[16]

With an increasing number of institutions and governments now implementing smoke-free policies covering areas other than enclosed public spaces, optimal understanding of the effectiveness of such policies in protecting children is essential. To establish this evidence base, we will undertake a comprehensive systematic review of available studies assessing the impact of policies covering outdoor areas or/and (semi)private places (whether indoor or outdoor) on children's TSE and respiratory health. We believe this work will be instrumental in informing effective policy-making, both (inter)nationally and locally, to further protect children from the adverse effects of tobacco smoke.

## METHODS
This protocol is reported according to the Preferred Reporting Items for Systematic Reviews and Meta-Analyses for Protocols (PRISMA-P) guideline for systematic review and meta-analysis protocols.[17]

### Patient and public involvement
Patients and the public were not involved in the design, development, conduct, reporting or dissemination of this study.

### Eligibility criteria
We will include studies based on predefined criteria as summarised in table 1 and detailed in the text below.[18]

#### Population
Studies are eligible when they include children between 0 and 16 years. Any study reporting (sub)populations in which at least 50% fits this age criterion will also be eligible. Some studies may have included specific subgroups, for example, children with asthma.[19] We will include all studies that meet the inclusion criteria, irrespective of such additional restrictions in the populations.

#### Interventions
We will include studies that evaluated smoke-free policies that were introduced at any governmental level (eg, cities, municipalities, regions or countries) or any institutional level (eg, school, hospital). The included policies might concern the following areas: indoor private places (eg, cars), indoor semiprivate places (eg, multiunit housing), outdoor (semi)private places (eg, shared gardens) and outdoor public places (eg, parks, school grounds, beaches, hospital grounds). We will exclude studies that solely evaluated policies covering enclosed public places (ie, 'traditional' smoke-free polices).

Smoke-free policies can be a part of a complex reform package.[20] For example, the above-described smoke-free

**Table 1** Inclusion criteria based on PICOS strategy

| PICOS | Inclusion criteria | Exclusion criteria |
|---|---|---|
| Population | 1. 0–16 years old<br>2. (Sub)populations in which at least 50% is under 17 years of age | Adult population without a distinct child subgroup |
| Intervention | Smoke-free policies instituted at any governmental or institutional level that restrict smoking in designated (semi)private places or/and any outdoor areas | Smoke-free policies covering only enclosed public places |
| Comparison | 1. A comparison population living in a location where no intervention was introduced/changed in the observational period<br>2. A comparison time period in which no intervention was introduced/changed | |
| Outcomes | I. Child TSE:<br>Primary outcome:<br>1. TSE in places covered by the policy (as reported by child and/or parent/primary caregiver)<br>Secondary outcomes:<br>1. TSE in places of which only some were covered by the policy or in unspecified places (as reported by child/parent/primary caregiver)<br>2. TSE in places not covered by the policy (as reported by child/parent/primary caregiver)<br>3. Cotinine or other specific biomarkers of TSE quantified in body fluids, hair, nails or on the skin<br>4. TSE assessed by wearable devices<br><br>II. Child health outcomes:<br>Primary outcomes:<br>1. Unplanned hospital attendance for wheezing/asthma<br>2. Unplanned hospital attendance for RTIs<br>Secondary outcomes:<br>1. General incidence of wheezing/asthma<br>2. General incidence of RTIs<br>3. OME<br>4. Chronic cough<br>5. FEV1, FVC, FEV1/FVC ratio | 1. Child TSE outcomes that are not specific to tobacco smoke (eg, $PM_{2.5}$ and CO)<br>2. Outcomes that do not necessarily imply a change in child TSE (eg, changes in tobacco smoke constituent level in a room)<br>3. Outcomes assessing smoking initiation/cessation or smoking behaviour (eg, among parents) |
| Study design | Included in the main analyses:<br>1. Randomised trials<br>2. Non-randomised trials<br>3. Interrupted time series<br>4. Controlled before–after studies<br>Included in sensitivity analyses:<br>1. Prospective cohort studies<br>2. Retrospective cohort studies<br>3. Uncontrolled before–after studies | |

Lung function represents FEV1, FVC, FEV1/FVC ratio.
CO, carbon monoxide; $FEV_1$, forced expiratory volume in one second; FVC, forced vital capacity; OME, otitis media with effusion; PICOS, Population, Intervention, Comparison, Outcomes, and Study design; $PM_{2.5}$, fine particulate matter; RTI, respiratory tract infection; TSE, tobacco smoke exposure.

policy might be introduced simultaneously with other tobacco control policies (eg, increased taxes on tobacco products). We will include all studies that estimate the effect of smoke-free policies covering (semi)private or/and outdoor places irrespective of whether other policies were introduced at the same time. If the effect of the smoke-free policy covering (semi)private or/and outdoor places has been disentangled from other interventions, we will extract the effect estimates from the analyses that most closely reflects that of this smoke-free policy.

### Comparison
We will include studies that estimate the counterfactual scenario (ie, no change in smoke-free policy implemented)

using any of the following comparators: (1) a comparison population where no change in the observed smoke-free policy occurred in the observational period or (2) a comparison time period in which no change in the observed smoke-free policy occurred.

## Outcomes

We will include studies assessing the impact of the policy on two types of outcomes: (1) indicators of TSE and (2) respiratory health outcomes. We have specified a wide range of outcomes to obtain a comprehensive overview of the entirety of available evidence on the topic. We will in our interpretation focus primarily on a small set of primary outcomes next to a larger set of secondary outcomes. Any study that reported step changes (immediate change in incidence) and/or slope changes (gradual change in incidence over time) in one or more of these outcomes will be eligible.

Studies that report on the following TSE indicators will be included:

Primary outcome:

1. TSE in places covered by the policy (as reported by the child/parent/primary caregiver).

Secondary outcomes:

1. TSE in places of which only some were covered by the policy, or in unspecified places (as reported by the child and/or parent/primary caregiver).
2. TSE in places not covered by the policy (as reported by the child/parent/primary caregiver).
3. Cotinine or other specific biomarkers of TSE quantified in body fluids, hair, nails or on the skin.
4. TSE assessed by wearable devices.

Note that the primary TSE outcome of interest assesses exposure specifically in places covered by the policy. That is, we aim to assess whether the policy was successful in reducing child TSE in those places that it was intended to cover. In addition, unintended effects of smoke-free policies on TSE have been described, both positive (spillover; eg, via increased adoption of voluntary home smoking bans following the implementation of smoke-free policy) and negative (displacement; eg, via increased smoking at home following smoke-free vehicle regulation).[14] In order to identify such effects, we will also identify studies that have assessed changes in TSE in places not covered by the policy.

Individual studies may report different estimates for TSE based on various definitions. We will prioritise child-reported TSE outcomes over parental or primary caregiver reported outcomes for studies reporting both. Furthermore, we will prioritise past-week TSE outcomes over longer or shorter recall periods. If past-week TSE is not available, longer recalls are prioritised over shorter recall periods.

The reduction in child TSE is anticipated to translate into child health benefits. Health outcomes are selected based on severity and responsiveness to changes in TSE within a reasonably short time frame (ie, within 1 year), these primarily being respiratory outcomes.[2 15] Accordingly, we will include studies that report the following outcomes:

Primary outcomes:

1. Unplanned hospital attendance for wheezing/asthma.
2. Unplanned hospital attendance for RTIs.

Secondary outcomes:

1. General incidence of wheezing/asthma.
2. General incidence of RTI.
3. Otitis media with effusion (OME).
4. Chronic cough.
5. Lung function (forced expiratory volume in one second (FEV1), forced vital capacity (FVC), FEV1/FVC ratio).

Unplanned hospital attendance may include acute presentations to the accident and emergency department as well as hospital admissions. For the secondary outcomes, respiratory diseases may be based on different case definitions. For studies reporting estimates according to various definitions, we will prioritise according to the following hierarchy: (1) based on physician diagnosis, (2) based on medication use (not applicable for OME) and (3) self-reported.

We will not include perinatal outcomes in this review as many of the policies for which we anticipate to identify evidence are specifically aimed at protecting children, not pregnant women (eg, smoke-free policies in playgrounds).

## Study designs

We will follow the guidelines of the Cochrane Effective Practice and Organisation of Care (EPOC) group to select eligible study designs.[21] EPOC established criteria for high-quality methodologies in research. These include: (1) randomised trials, (2) non-randomised trials, (3) interrupted time series and (4) controlled before–after studies. We will include studies that satisfy these criteria in our main analysis. Based on previous experiences, this restriction may sometimes lead to a lower number of studies being eligible. In order to properly assess the available evidence in such instances, we will conduct sensitivity analyses by relaxing the inclusion criteria.[2 22] For the sensitivity analyses, we will additionally include (1) prospective cohort studies, (2) retrospective cohort studies and (3) uncontrolled before–after studies.

## Further restrictions

There will be no restriction on publication date, the timeframe of the study or the length of follow-up after the introduction of the policy. All databases will be searched from their date of inception, and we will update the search to supplement our review just before publication. We will include every study that meets the above-mentioned criteria regardless of the language. Studies published in languages other than English will be translated. Google Translate will be used for this purpose, and we will consult with a translator if necessary.

## Report type

Our systematic review will include published studies in scientific journals as well as in the 'grey literature'

(ie, documents that are circulated in non-commercial academic channels or/and not indexed by major data-bases). Only full reports will be included in our review. Studies for which only an abstract was published will not be included since risk of bias for these studies cannot be adequately assessed. In these cases, however, we will contact the authors to ask if a full report is available.

## Information sources

We will search for potential studies in the following electronic databases: (1) Embase.com, (2) Medline Ovid, (3) Web of Science, (4) PsycINFO Ovid, (5) CINAHL EBSCOhost, (6) Google Scholar, (7) IndMED, (8) KoreaMed, (9) EconLit, (10) WHO Global Health Library (including African Index Medicus, Latin America and the Caribbean Literature on Health Sciences, Index Medicus for the Eastern Mediter-ranean Region, Index Medicus for South-East Asia Region, Western Pacific Region Index Medicus, (11) WHO Library Database, (12) Scientific Electronic Library Online and (13) Paediatric Economic Database Evaluation.

## Search strategy

First, we will search for potentially relevant studies based on a search strategy that is the combination of Medical Subject Headings (MeSH) terms and free text search. Our research team, including a librarian who is specialised in search strategy optimisation, has developed this search strategy. Search terms were tailored to each database. By means of an example, search terms used for Embase.com database are available in online supplemental appendix. Search terms included four parts: (1) terms to identify smoke-free policies; (2) terms that identify children as the target population; (3) terms to identify asthma, wheezing, respiratory diseases, OME, chronic cough, lung function or TSE as outcome and (4) terms that exclude confer-ence abstracts, letter to the editors, notes and editorials.

Second, we will search for additional studies by screening reference lists of included studies and their cita-tions through Google Scholar. Moreover, we will contact with experts in the field to identify additional studies that may have been missed and any relevant ongoing or unpublished studies. Finally, we will update our search to add the most recent publications just before submitting the review for publication.

## Study records

### Data management

We will extract all records identified by the different sources into an EndNote Library, and use this software to automat-ically deduplicate the collected records according to the method previously described.[23] Subsequently, we will manu-ally identify and remove any duplicates that remain. At this stage, duplicates will be identified based on overlaps in the authors' names, the titles of the publication, the observed populations, the constructions of the treatment group and control groups, sample sizes and the reported outcomes. In our final report, we will note the number of duplicates in the PRISMA flow diagram.

## Selection process

After the deduplication process, we will select eligible papers from the remaining unique records. At the first step, two independent researchers will screen titles and the abstracts for eligibility. We will then obtain the full-text reports of studies that may fit eligibility criteria based on this assessment. At the next step, two independent researchers will assess eligibility based on the full texts. Any discrepancies will be resolved after discussion with a third researcher. The involved researchers will not be blinded to information about the articles (eg, authors' names and affiliations) at any stage.

## Data collection process

Two researchers will independently extract data from the included studies to a customised data extraction form developed a priori that has been piloted using six eligible studies. After finishing the data extraction, they will confer their results with each other and create one final file. Again, any discrepancies will be resolved after discussion with a third researcher. We will contact authors for any relevant missing data.

In case overlapping populations are analysed in multiple studies, we will include according to the following hierarchy the study that (1) has the lowest risk of bias (see risk-of-bias assessment below), (2) evaluates the most comprehensive policy or (3) incorporates the largest sample size. In case one study reports multiple effect estimates for overlapping populations, we will select according to the following hierarchy the one derived from (1) the most adjusted model, (2) the longest obser-vation period, (3) the most comprehensive policy or (4) the largest treatment group.

## Data items

From eligible studies, we will extract the following items: (1) the first author's name, (2) his or her affiliation, (3) publication year, (4) type of publication (journal, book, dissertation, etc), (5) access information (URL or doi), (6) study design, (7) observational period (the beginning and end dates), (8) exact places covered by the policy, (9) whether the policy covers outdoor or indoor places, (10) whether the policy covers public or (semi)private, places, (11) timing of policy, (12) institution/govern-ment that initiated the policy, (13) country/location where the policy had been implemented, (14) compli-ance with the policy, (15) enforcement of the policy, (16) eligibility criteria for inclusion, (17) description of control group(s), (18) population at risk, (19) number of participants/events, (20) control and treatment group size, (21) characteristics of the population and the treat-ment groups (eg, age, gender and socioeconomic status), (22) data source(s) used for the study, (23) definition(s) of outcome measure(s), (24) controlled confounders (if applicable), (25) applied statistical techniques to draw inference, (26) number of clusters (if applicable), (27) cluster size (if applicable), (28) whether the results were adjusted for clustering (if clustered study), (29) whether

the intraclass correlation coefficient is reported and what it is (if clustered study), (30) the number of drop-outs/missing values, (31) techniques for handling missing values, (32) preintervention population at risk(n)/events(n)/rates (%) of outcome variable(s), (33) postintervention population at risk(n)/events(n) rates (%) of outcome variable(s), (34) association between smoke-free policy and outcome(s) (coefficients, CIs and p values), (35) any unintended effects (eg, TSE displacement), (36) proportion of children exposed to TSE in the location covered by the smoke-free policy, before and after the implementation of the policy, (37) risk ratios (RRs) of respiratory disease for different levels of TSE according to specific locations covered by the smoke-free policy, (38) bias assessment (see section 'Risk of bias assessment'), (39) elements supporting causal inference (see section 'Elements supporting causal inference'), (40) any conflict of interest reported by the authors and (41) the funding source(s).

### Outcomes and prioritisation

As previously detailed, we will identify studies evaluating the effects of smoke-free policies on children's (1) TSE and (2) health outcomes. See the description of outcomes and our prioritisation in the 'Eligibly criteria' section.

### Risk of bias assessment

For each individual result, we will assess risk of bias using standardised assessment tools. For randomised trials, we will use the quality assessment tool for quantitative studies, developed by EPOC.[24] For non-randomised studies, we will use the Risk Of Bias In Non-randomised Studies of Interventions (ROBINS-I) tool.[25] The ROBINS-I tool evaluates biases in the following domains: (1) bias due to confounding (online supplemental appendix table 1), (2) bias in selection of participants into the study, (3) bias in classification of interventions, (4) bias due to deviations from intended interventions (see online supplemental appendix), (5) bias due to missing data, (6) bias in measurement of outcomes and (7) bias in selection of the reported result. For each domain, we will rate the risk of bias on a 4-point scale ranging from critical risk of bias to low risk of bias (ie, bias is comparable to a well-performed randomised trial). The ROBINS-I tool will be followed to give an overall estimate of the risk of bias. The risk-of-bias assessments will be conducted independently by two researchers and a third reviewer will be consulted to resolve disagreements.

### Elements supporting causal inference

Based on information reported in the included studies, we will reflect on elements that may support causal inference and the robustness of the evidence using the UK Medical Research Council guidance on natural experiments.[26] First, we will report whether the effect estimates were robust when using alternative comparison groups (eg, the use of different populations living in distinct geographical areas as a comparison). Second, we will

report whether a study estimated whether the smoke-free policy also had an 'effect' on neutral outcomes that were not expected to change as a consequence of the intervention (eg, hospital admission for appendicitis). Finally, we will report from the included studies any additional information from complementing research methodologies regarding possible underlying mechanisms supporting the quantitative study findings. These assessments will be based on the quantitative and qualitative information presented in the individual study reports. We anticipate that this information cannot be uniformised across studies, and as such will be reported narratively.

### Data synthesis
#### Synthetisation of data

Obtaining comparable data is essential to facilitate meta-analysis. Studies should be sufficiently similar in terms of their designs and the definitions of their measurements. We will uniformise outcome assessments before conducting data analysis. First, if continuous outcomes appear on different scales across studies, we will use standardised mean differences. If continuous outcomes appear on the same scale, weighted mean differences will be used. Second, for dichotomous outcomes, we will extract relative risks. If only absolute risks are presented, then we will calculate relative risks. Third, we will express the effects on dichotomous outcomes using RR. If RR was not reported, we will contact the authors to provide it to us. If the requested data were not obtained, we will convert OR to RR using the following formula:

$$RR = \frac{OR}{(1-PEER)+(PEER \times OR)}$$

, where refers to the patient-expected event rate in the control group.[27] In case PEER is not available, we will approximate this measure using the overall event rate across the entire study population. We will use incidence rate ratios instead of RR for outcomes that could occur repeatedly with the same individual (ie, hospitalisation). To make different TSE measures comparable, we will convert them into a binary variable with a value of 1 if the child was exposed to (detectable amounts of) TSE and 0 if unexposed to (detectable amounts of) TSE. Finally, two researchers will check independently whether the definitions of intervention, population and outcomes are sufficiently comparable across the selected studies to allow meta-analysis.

The eligible studies are likely to be heterogeneous in various important characteristics that will be taken into account in the data synthesis. Therefore, we will apply random effects models (as opposed to fixed-effect models) for data analysis of two or more studies with similar policies and outcome measure. Furthermore, it might be needed to take into account the dependency of observations, for example, if multiple effect estimates for similar policies across various regions were provided within a single study. A three-level meta-analysis will be considered if multiple estimations are extracted from the same study. The decision to perform a two-level or

three-level meta-analysis will be based on the model performance using a one-sided log-likelihood-ratio test.[28 29] We will separately pool step changes (immediate changes) and slope changes (gradual changes) for each outcome in different models. If unspecified, the result will be considered a step change. We will conduct separate meta-analyses for various subtypes of smoke-free policies, according to the locations that they cover. Thus, we will separately pool studies for (1) smoke-free policies covering indoor private places (eg, cars), (2) smoke-free policies covering indoor semiprivate places (eg, multi-unit housing), (3) smoke-free policies covering outdoor private or semiprivate spaces (eg, shared gardens) and (4) smoke-free policies covering outdoor public places (eg, parks, school grounds, beaches and hospital grounds). Finally, regarding TSE, we will pool separate models to assess unintended effects of the policy in places not covered by the policy.

The estimations of our meta-analysis will be displayed in forest plots. Heterogeneity will be quantified by the $I^2$ statistic for each pooled model.

We anticipate that most studies will have assessed the impact of novel smoke-free policies on TSE rather than on health outcomes. As the relationship between TSE and respiratory outcomes among children is well established,[30 31] we will use data from studies estimating the impact of novel smoke-free policies on child TSE to quantify the potential impact of these changes on our primary health outcomes. Thus, we will perform a health impact assessment and quantify the proportion of children with respiratory disease that can be prevented by the change in TSE following smoke-free policies. First, we will extract the RR of respiratory disease for those children who were exposed to TSE versus those who were not either from the studies included in our systematic review or from previous meta-analyses that quantified this association.[30 31] Preferably, this RR will be retrieved for the specific locations that are covered by the smoke-free policy and for distinct respiratory diseases. Second, based on the RRs and the difference in the proportion of children exposed to TSE before and after the policy implementation, we will calculate two population attributable fractions (PAFs). One quantifies the proportion of each outcome that can be attributed to TSE before the smoke-free policy and the other after the policy's implementation. The difference of these PAFs will give us the potential impact fraction capturing the change in the outcomes when the TSE level changes following a novel smoke-free policy's implementation.

### Sensitivity analyses
Three sets of sensitivity analyses will be conducted. First, excluding results with a higher risk of bias might lead to different estimations. Thus, we will conduct meta-regression analyses using the overall risk of bias based on the ROBINS-I assessment to test if effect sizes differ according to the overall risk of bias. Second, the question may arise whether a less restrictive selection strategy

regarding study design would lead to different results. We will check for the robustness of our results by additionally including studies applying not only methodologies considered as the state of the art in the Cochrane EPOC guidelines, but also study designs with a larger risk of bias.[21] Third, we will test the sensitivity of our results to the selection of studies when overlapping populations are analysed in multiple studies.

### Subgroup analyses
We will conduct subgroup analyses to evaluate whether smoke-free policies have differential effects among certain subgroups. To assess whether policies introduced in high-income countries have a different effect than policies introduced in low-income or middle-income countries, we will perform meta-regression based on the World Bank classification.[32] Second, previous studies have shown that smoke-free policies have a different effect according to individuals' SES.[22 33–35] We will extract information from subgroup analyses by SES indicators where possible and perform separate meta-analyses for high and low SES groups.

### Meta-bias assessment
In each meta-analysis, we will assess the extent of publication bias via funnel plots when at least 10 studies are available. Furthermore, we will look for unpublished studies for which a protocol was registered, and we will contact the authors of these studies to obtain their results. Subsequently, we will compare the findings of these unpublished studies with those of the published reports. Finally, we will look for published reports that followed a registered protocol. We will compare these protocols with the final papers to check whether selective reporting was present at any stage.

### Confidence in cumulative estimate
We will discuss the strength of the evidence on the association between implementation of smoke-free policies covering (semi)private or/and outdoor places and the TSE and child respiratory health outcomes based on the effect sizes, the variances of the estimated effects, the estimated level of heterogeneity between studies, sensitivity analyses, subgroup analyses, risk of bias, elements supporting causal inference and publication bias.

### ETHICS AND DISSEMINATION
Ethical approval will not be required because we will only be using aggregate-level published data from previous studies. We expect to complete the study until September 2020, and before submitting for publication, we will update our search and include any additional eligible papers that may be identified. We will publish our results in a peer-reviewed international journal and report the research findings according to the PRISMA guideline.[36] Finally, we will disseminate

our results to the public and policy-makers following the acceptance of our paper (eg, actively seek media attention and interaction with policy makers and other relevant stakeholders).

**Author affiliations**
[1]Department of Paediatrics, Erasmus MC - Sophia Children's Hospital, University Medical Centre Rotterdam, Rotterdam, Netherlands
[2]Department of Public Health, Erasmus MC, University Medical Centre Rotterdam, Rotterdam, Netherlands
[3]Usher Institute, The University of Edinburgh, Edinburgh, United Kingdom
[4]Brigham and Women's Hospital, Harvard Medical School, Boston, Massachusetts, USA
[5]Public Health Policy Evaluation Unit, School of Public Health, Imperial College London, London, United Kingdom
[6]Medical Library, Erasmus MC, University Medical Centre Rotterdam, Rotterdam, Netherlands

**Acknowledgements** We thank Lauren Westenberg, Brigit Toebes and Martine Bouman for providing input.

**Contributors** JVB secured funding for this work. All authors contributed to the conceptualisation of the review and developed the methods. MR, FM and JVB drafted the manuscript and AS, CM, WMB, AB and FJVL provided feedback on manuscript drafts. JVB and WMB developed the search strategy. All authors approved the manuscript.

**Funding** This study is funded by a project grant from the Dutch Heart Foundation, Lung Foundation Netherlands, Dutch Cancer Society, Dutch Diabetes Research Foundation and the Netherlands Thrombosis Foundation. The Public Health Policy Evaluation Unit at Imperial College London is grateful for the support of the NIHR School of Public Health Research.

**Competing interests** None declared.

**Patient and public involvement** Patients and/or the public were not involved in the design, or conduct, or reporting, or dissemination plans of this research.

**Patient consent for publication** Not required.

**Provenance and peer review** Not commissioned; externally peer reviewed.

**ORCID iDs**
Márta K Radó http://orcid.org/0000-0002-1676-5951
Famke JM Mölenberg http://orcid.org/0000-0002-5305-9730
Christopher Millett http://orcid.org/0000-0002-0793-9884
Wichor M Bramer http://orcid.org/0000-0003-2681-9180
Alex Burdorf http://orcid.org/0000-0003-3129-2862
Jasper V Been http://orcid.org/0000-0002-4907-6466

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
