## [Reviewer comments · BMJ Open]

ARTICLE DETAILS

TITLE (PROVISIONAL)	Impact of expanding smoke-free policies beyond enclosed public places and workplaces on children's tobacco smoke exposure and respiratory health: protocol for a systematic review and meta-analysis
AUTHORS	Rado, Márta; Mölenberg, Famke; Sheikh, Aziz; Millett, Christopher; Bramer, Wichor M.; Burdorf, Alex; Van Lenthe, Frank J.; Been, JV

VERSION 1 – REVIEW

REVIEWER	Vicki Myers Haifa University Israel
REVIEW RETURNED	07-Apr-2020

GENERAL COMMENTS	The study protocol seems sound and well-planned, and the review will address an important question regarding effectiveness of smokefree policies in outdoor and semi-private areas on exposure and child health. I look forward to seeing the results of the systematic review. Two comments: 1. According to the journal requirements, dates of the study should be included (both when it will be conducted, and which years will be included in the data search).2. As mentioned it is likely that studies will be difficult to compare, presenting different outcomes, and that there may be a small number of relevant studies for each of the numerous mentioned outcomes, limiting the power of meta-analysis. Depending on available data the authors may need to focus on fewer outcomes.
---

REVIEWER	Leah Stevenson James Cook University, Australia
REVIEW RETURNED	07-May-2020

GENERAL COMMENTS	On the onset, I found your protocol to investigate the impact of novel smoke-free policies an interesting topic. The protocol is thorough; however, a few points need further explanation to improve clarity. The review that will be conducted from this protocol will be of interest to me, and I look forward to reading the systematic review in the future. Introduction 1. Paragraph 1 (2 -7). Every second child worldwide is very broad – the rates of TSE will be significantly different between countries/regions/states within countries. E.g. Provide examples of how general smoke-free policies can influence TSE, SHS, THS, which have created different outcomes for different regions.
--

	2. 11-12- where the statistical measures metanalysis results from the systematic review or individual results from separate studies- be specific. 3. 13-14- Please confirm if the results you show above are from areas with smoke-free policies. 4. 33 Spell check 'unknow' 5. 33-34. Please explain the negative and positive concepts for TSE exposure more clearly 6. 39- Please state if this will be qualitative, quantitative or if the review covers both. It is stated that qualitative articles will be explored in section 'Elements of supporting causal inference' however there isn't clear information about qualitative articles being used in the review in your aim or methods (see comments below). Methods 7. Your inclusion and exclusion studies in your Table 1 doesn't show the quality assessment tool you will be using qualitative studies. 8. Please provide details of how you are going to analyse the qualitative studies. How are the 'mechanisms' going to be identified?
--	---

VERSION 1 – AUTHOR RESPONSE

Response to Referee 1:

The study protocol seems sound and well-planned, and the review will address an important question regarding effectiveness of smokefree policies in outdoor and semi-private areas on exposure and child health. I look forward to seeing the results of the systematic review.

Two comments:

1. According to the journal requirements, dates of the study should be included (both when it will be conducted, and which years will be included in the data search).

Response: We have added the expected dates of conducting the review in the following section:

Line 393-394: "We expect to complete the study until September 2020 and before submitting for publication we will update our search and include any additional eligible papers that may be identified."

Line 207-208: "All databases will be searched from their date of inception, and we will update the search to supplement our review just before publication."

2. As mentioned it is likely that studies will be difficult to compare, presenting different outcomes, and that there may be a small number of relevant studies for each of the numerous mentioned outcomes, limiting the power of meta-analysis. Depending on available data the authors may need to focus on fewer outcomes.

Response: We set up this study to evaluate the effectiveness of novel smoke-free policies on children's tobacco smoke exposure (TSE) and respiratory health. We aim to include all relevant outcomes known from previous reviews to make our review as comprehensive as possible and not miss any relevant evidence given the novelty of this research area. According to the reviewer's suggestion, we focus on a small number of outcomes in our primary analyses by making a clear distinction between primary and secondary outcomes. To address the reviewer's concern regarding the potentially low number of expected studies available for meta-analysis, we decided to lower our threshold for conducting a meta-analysis from three to two available studies, according to Cochrane criteria.¹

Line 150-152: "We have specified a wide range of outcomes to obtain a comprehensive overview of the entirety of available evidence on the topic. We will in our interpretation focus primarily on a small set of primary outcomes next to a larger set of secondary outcomes."

Line 333-334: "Therefore we will apply random-effects models (as opposed to fixed-effect) for data analysis of two or more studies with similar policies and outcome measure."

Response to Referee 2:

On the onset, I found your protocol to investigate the impact of novel smoke-free policies an interesting topic. The protocol is thorough; however, a few points need further explanation to improve clarity. The review that will be conducted from this protocol will be of interest to me, and I look forward to reading the systematic review in the future.

Introduction

1. Paragraph 1 (2 -7). Every second child worldwide is very broad – the rates of TSE will be significantly different between countries/regions/states within countries. E.g. Provide examples of how general smoke-free policies can influence TSE, SHS, THS, which have created different outcomes for different regions.

Response: We added information on the local differences in the effects of smoke-free policies on adverse health outcomes. We also discuss the differences between high and low-income countries.

Line 68-69: "Tobacco smoke exposure (TSE) is an important cause of adverse respiratory health outcomes in children worldwide."

Line 83-85: "Although in low and middle-income countries high background air pollution, poor economic conditions, and low awareness of tobacco-related harm might obscure the positive effects of smoke-free legislation, evidence suggests that smoke-free policies can have a similarly positive impact in these countries as in high-income countries.^{3 4}"

Line 105-106: "Finally, effectiveness of smoke-free policies is likely to vary according to local rates of smoking and children's SHS exposure, and comprehensiveness of the policy as well as compliance and enforcement.²"

2. 11-12- where the statistical measures meta-analysis results from the systematic review or individual results from separate studies- be specific.

Response: We have specified that the results were statistics from meta-analyses.

Line 75-78: "Meta-analysis showed that these policies were followed by a 9.8% (95% confidence interval [CI] 3.0 – 16.6) reduction in hospital attendance for asthma exacerbations, and an 18.5% (95% CI 4.2– 32.8) reduction in hospital attendance for lower RTIs."

3. 13-14- Please confirm if the results you show above are from areas with smoke-free policies.

Response: We have specified that these results are from areas with smoke-free policies.

Line 79-80: "These health impacts of smoke-free policies are likely mediated through a reduction in SHS and potentially also in THS."

4. 33 Spell check 'unknow'

Response: We have corrected the typo.

5. 33-34. Please explain the negative and positive concepts for TSE exposure more clearly

Response: We have provided a better explanation of these negative and positive unintended effects.

Line 99-103: "Further, evidence from various countries indicates that smoke-free policies covering public enclosed places and workplaces are followed by reductions in TSE even in areas not covered by the policy through norm spreading. Whether novel smoke-free policies also have an impact on TSE in places not covered by the policy is unclear. Theoretically, there may be a reduction (via norm spreading) in such places but there may also be displacement of smoking to areas not covered by the policy."

6. 39- Please state if this will be qualitative, quantitative or if the review covers both. It is stated that qualitative articles will be explored in section 'Elements of supporting causal inference' however there isn't clear information about qualitative articles being used in the review in your aim or methods (see comments below).

Response: This review will only include quantitative studies reporting on the effect of novel smoke-free policies on TSE and/or respiratory health in children. From these studies, we will however also extract additional elements, either quantitative or qualitative, that may provide further insights on the mechanisms of how the policies under evaluation may have impacted the outcomes. This has been clarified in the revised text:

Line 298-299: "Based on information reported in the included studies, we will reflect on elements that may support causal inference and the robustness of the evidence using the UK Medical Research Council guidance on natural experiments."

Line 305-309: "Finally, we will report from the included studies any additional information from complementing research methodologies regarding possible underlying mechanisms supporting the quantitative study findings. These assessments will be based on the quantitative and qualitative information presented in the individual study reports. We anticipate that this information cannot be uniformised across studies, and as such will be reported narratively."

Methods

7. Your inclusion and exclusion studies in your Table 1 doesn't show the quality assessment tool you will be using qualitative studies.

Response: As described above, we will not be including qualitative studies. We will however from quantitative studies report supporting information, either quantitatively or qualitatively, that may support causal inference.

8. Please provide details of how you are going to analyse the qualitative studies. How are the 'mechanisms' going to be identified?

Response: This systematic review will include quantitative studies only. From the included studies we will extract additional elements (either qualitative or quantitative evidence) that may support causal inference. The additional elements will be summarised in tables and the manuscript text. We anticipate that this information will not be uniform across studies. Therefore, we will narratively describe if additional elements reported in a particular study support its quantitative findings.

Line 305-309: "Finally, we will report from the included studies any additional information from complementing research methodologies regarding possible underlying mechanisms supporting the quantitative study findings. These assessments will be based on the quantitative and qualitative information presented in the individual study reports. We anticipate that this information cannot be uniformised across studies, and as such will be reported narratively."

In addition to the changes suggested by the reviewers, we have added a bit more detail on our planned analytical approach in some places in the manuscript:

Line 335-339: "Furthermore, it might be needed to take into account the dependency of observations, for example, if multiple effect estimates for similar policies across various regions were provided within a single study. A three-level meta-analysis will be considered if multiple estimations are extracted from the same study. The decision to perform a 2-level or 3-level meta-analysis will be based on the model performance using a one-sided log-likelihood-ratio test.^{5,6}"

Reference

1. Higgins JP, Green S. Cochrane Handbook for Systematic Reviews of Interventions Version 6. 2018
2. World Health Organization. Report on the global tobacco epidemic: offer help to quit tobacco use: executive summary. 2019
3. Hone T, Szklo AS, Filippidis FT, et al. Smoke-free legislation and neonatal and infant mortality in Brazil: longitudinal quasi-experimental study. *Tobacco Control* 2020(29):312-19.

4. Byron MJ, Cohen JE, Frattaroli S, et al. Implementing smoke-free policies in low-and middle-income countries: A brief review and research agenda. *Tobacco induced diseases* 2019;17
5. Assink M, Wibbelink CJ. Fitting three-level meta-analytic models in R: A step-by-step tutorial. *The Quantitative Methods for Psychology* 2016;12(3):154-74.
6. Cheung MW-L. Modeling dependent effect sizes with three-level meta-analyses: a structural equation modeling approach. *Psychological Methods* 2014;19(2):211.